# UNDERSTANDING SKILL ADAPTATION IN TRANSFORMERS USING SPARSE AUTOENCODERS: CHESS AS A MODEL SYSTEM

## ABSTRACT

Understanding how skill shapes decision-making in complex environments is a challenging problem in AI interpretability. We investigate this question by applying Sparse Autoencoders (SAEs) to the internal representations of Maia-2, a human-like chess model that simulates human play across varying skill levels. Maia-2 incorporates a skill-aware transformer that integrates position features with categorical skill inputs, capturing nuanced relationships between player expertise and move selection. By training SAEs on these modulated representations, we identify latent features that are specifically activated by interpretable chess concepts. We then apply mediated interventions with targeted SAE features to effectively elicit both higher and lower skill play in relevant sets of chess positions. Our findings suggest that SAE features can help shed light on how skill-specific information is encoded within a model to produce human-like behavior, and that these insights can be applied to steer the model's performance on downstream tasks. Our work is available at `https://anonymous.4open.science/r/chess-sae-3C06/`

## 1 INTRODUCTION

A flurry of recent work has deepened our understanding of how models like transformers function. Applying mechanistic interpretability techniques to either language models, custom artificial intelligence systems, or some combination of the two, researchers have established that mechanistic interpretability can reveal the process by which AI models acquire human-like knowledge in complex domains (McGrath et al., 2022), that transformers can learn look-ahead (Jenner et al., 2024), and that language models develop emergent world models that can be productively edited (Gurnee & Tegmark, 2023).

Research in Large Language Models (LLMs) have demonstrated their ability to adapt to specified skill or expertise levels. Well-trained models can not only perform complex tasks, but can do so at levels ranging from beginner to master. This capability, often referred to as *role-playing*, allows users to elicit responses from LLMs that simulate different levels of expertise or personas (Wei et al., 2022b; Brown, 2020; Shanahan et al., 2023). For example, people routinely condition large language models in their prompts by explicitly telling them to, e.g., "Act as if you were a world-class programmer", or "Pretend you don't know much about trigonometry", and they do a convincing job in response.

However, it remains unclear how models like transformers successfully model different skill levels. How can the LLMs do this in the domain of language? Some researchers have hypothesized that this ability stems from the models' exposure to diverse writing styles and expertise levels during training (Wang et al., 2022). Others suggest that it may be an emergent property of the models' large-scale knowledge integration (Wei et al., 2022a). Despite these hypotheses, there is a critical gap in the understanding of internal representations and transformations that enable LLMs to modulate their outputs across different skill levels.

On the other hand, understanding the mechanisms by which models capture skill would have several important benefits. First, we could better understand how models adapt their output to suit a specified level, which would be practically useful for designing applications that depend on this ability,

including catching errors the model may make or inconsistencies that may arise. Second, and perhaps most interestingly, if we knew the mechanisms by which models capture skill, we could take a meaningful step towards truly algorithmic coaching. A great teacher understands both where we are at and where we want to be, and how to get us from A to B. A deeper understanding of how models capture these different levels would crystallize the knowledge required to build AI-powered tutors that can make personalized recommendations to transform our skill from where it is today to where it could be tomorrow.

Unlike the open-ended domain of language, however, chess provides a well-defined space with clear metrics for skill levels. Recent studies showcase the inroad of applying mechanistic interpretability techniques in understanding the internal mechanism of chess models (Jenner et al., 2024; Karvonen et al., 2024). Unlike Karvonen, we do not seek to measure the quality of sparse autoencoders (SAEs) against a pre-defined set of chess concepts related to the state of the game. Instead, we use SAEs as our tool to understand how Maia-2, the state-of-the-art human move prediction model in chess, adapts its predictions to the skill level of the target player it is modeling. The exploration of skill adaptation in Maia-2 model offers a unique opportunity to study this phenomenon in a more controlled environment. We train SAEs on the modulated representations of Maia-2 to identify latent features that learn well on interpretable chess concepts. We then apply mediated interventions with SAE features to effectively elicit both higher and lower skill play in specific sets of chess positions.

Our key findings demonstrate that the features of our Maia-2 SAEs can effectively capture and manipulate skill-sensitive information within the model. These results not only provide insights into how skill-specific information is encoded within the model to produce human-like behavior but also demonstrate that these insights can be applied to adjust the model's performance on specific downstream tasks.

## 2 BACKGROUND

The intersection of artificial intelligence and chess has been a fertile ground for research in human-AI alignment and interpretability. This section provides an overview of key developments in chess AI, human-like chess models, and interpretability techniques that form the foundation for our work.

### 2.1 CHESS AND AI

Chess has long served as a benchmark for artificial intelligence, with significant milestones including IBM's Deep Blue defeating world champion Garry Kasparov in 1997 (Hsu, 1999). Modern chess engines like Stockfish (Romstad et al., 2023) use sophisticated heuristic search techniques, while approaches like AlphaZero (Silver et al., 2018) leverage deep learning and self-play to achieve superhuman performance. Recent studies also propose training transformer-based models on textual chess annotations with games (Feng et al., 2024), and treating chess board positions as discrete tokens (Leela Chess Zero, 2024).

**human-AI alignment in Chess.** However, the superhuman chess models often play in ways that are difficult for humans to understand or learn from. This gap led to the development of human-like chess models, aimed at capturing and predicting human behavior in chess. The Maia series of models (McIlroy-Young et al., 2020; Tang et al., 2024) represent significant advancements in this direction, with Maia-2 introducing a unified architecture capable of modeling chess play across various skill levels with coherence.

### 2.2 MECHANISTIC INTERPRETABILITY IN CHESS MODELS

**Linear Probes.** A direct way of extracting useful knowledge from superhuman systems is to train probes over model internals in a human representation space. Without any prior human guidance, evidence of human chess concepts learned by AlphaZero is found and quantitatively measured by linear probes (McGrath et al., 2022).

**Dictionary learning with Sparse Autoencoders.** Sparse Autoencoders (SAEs) have emerged as a powerful tool for interpreting the internal representations of neural networks. Recent work by

Cunningham et al. (2023) and Templeton et al. (2024) has shown that SAEs can effectively learn interpretable and monosemantic features from language models. In the domain of chess, Karvonen et al. (2024) demonstrated that SAEs can learn features corresponding to strategic chess concepts and board state properties.

**Activation Patching.** Activation patching is a technique for measuring the causal importance of specific model components. Jenner et al. (2024) applied this method to study learned look-ahead in Leela Chess Zero. By replacing activations from a clean board state with those from carefully chosen corrupted states, they identified model components where future move information is stored.

## 3 PRELIMINARIES

### 3.1 MAIA-2 ARCHITECTURE

The Maia models are a series of chess AI designed to predict human moves with high accuracy across various skill levels. The original Maia-1 (McIlroy-Young et al., 2020) consists of nine separate models, each trained on games from players within a specific rating range. While effective, this approach led to inconsistencies across skill levels and limited the models' ability to adapt to diverse player matchups. Maia-2 addresses these limitations by introducing a unified architecture capable of coherently modeling chess play across the spectrum of skill levels(Tang et al., 2024). It employs a novel skill-aware attention mechanism to dynamically integrate player skill information with chess positions.

Maia-2 consists of lower layers of Resnet-based backbone network and transformer-based skill-aware blocks above. The model takes as input a chess position, represented as a multi-channel tensor $\mathbf{P}_{\text{input}} \in \mathbb{R}^{C_{\text{board}} \times 8 \times 8}$, along with skill levels of both the active player and the opponent. These skill levels are encoded into embeddings $\mathbf{e}_a, \mathbf{e}_o \in \mathbb{R}^{d_s}$ using a skill level encoder.

The ResNet-based backbone processes $\mathbf{P}_{\text{input}}$ through convolutional blocks to produce an encoded representation $\mathbf{P}_{\text{encoded}} \in \mathbb{R}^{C_{\text{patch}} \times 8 \times 8}$. This representation undergoes channel-wise patching and linear transformation as the inputs, denoted as $P$, to the residual flows of the transformer blocks of Maia-2 as following:

$$P_{\text{patched}} = \text{Patching}(P_{\text{encoded}}) \in \mathbb{R}^{C_{\text{patch}} \times 64} \tag{1}$$

$$P = P_{\text{patched}}\mathbf{W} + \mathbf{b} \in \mathbb{R}^{C_{\text{patch}} \times d_{\text{att}}}, \tag{2}$$

In its transformer blocks, Maia-2 performs the skill-aware attention mechanism to fuse skill embeddings with the position representations. The queries $Q_k$, keys $K_k$, and values $V_k$ are transformed with their respective embedding matrix from $P$. For each attention head $k$, the queries injected with skill information $Q_k^*$ are computed as:

$$Q_k^* = Q_k + (\mathbf{e}_a \oplus \mathbf{e}_o)\mathbf{W}^* \tag{3}$$

where $\oplus$ denotes concatenation. The attention output is then calculated as:

$$\mathbf{h}_k = \text{softmax}\left(\frac{\mathbf{Q}_k^*\mathbf{K}_k^T}{\sqrt{d_k}}\right)\mathbf{V}_k \tag{4}$$

$$P_{\text{att}} = \sigma((h_1 \oplus h_2 \oplus \ldots \oplus h_h)\mathbf{W}^O) \tag{5}$$

We then apply the vanilla ViT's feed-forward network and add & norm components upon $P_{att}$ to obtain the output of each skill-aware block and add it back to the model's residual stream. The model's output includes a policy head for move prediction, a value head for game outcome prediction, and an auxiliary information head.

### 3.2 SPARSE AUTOENCODERS

A sparse autoencoder (SAE) is designed as a one-layer network that decomposes and reconstructs the internal representations from a model, while enforcing sparsity in its hidden layer activations.

Given an input vector $\mathbf{x} \in \mathbb{R}^{d_{\text{in}}}$, where $d_{\text{in}}$ is the dimension of the internal representation, the SAE performs encoding and decoding operations as follows:

$$\mathbf{c} = \sigma(\mathbf{W}_{\text{enc}}\mathbf{x} + \mathbf{b}_{\text{enc}}) \tag{6}$$

$$\hat{\mathbf{x}} = \mathbf{W}_{\text{dec}}\mathbf{c} + \mathbf{b}_{\text{dec}} \tag{7}$$

where $\mathbf{W}_{\text{enc}} \in \mathbb{R}^{d_{\text{hid}} \times d_{\text{in}}}$ and $\mathbf{W}_{\text{dec}} \in \mathbb{R}^{d_{\text{in}} \times d_{\text{hid}}}$ are the encoding and decoding weight matrices respectively, and $\mathbf{b}_{\text{enc}} \in \mathbb{R}^{d_{\text{hid}}}$ and $\mathbf{b}_{\text{dec}} \in \mathbb{R}^{d_{\text{in}}}$ are the encoding and decoding bias vectors. Typically, the dimension of the hidden layer $d_{\text{hid}}$ is chosen such that $d_{\text{hid}} > d_{\text{in}}$. $\sigma(\cdot)$ is an activation function. The SAE is trained to minimize a loss function that combines reconstruction loss and a sparsity penalty:

$$\mathcal{L}(\mathbf{x}) = \|\mathbf{x} - \hat{\mathbf{x}}\|_2^2 + \alpha\|\mathbf{c}\|_1 \tag{8}$$

where $\alpha$ is a hyperparameter controlling the sparsity of the hidden layer activations. The columns of $\mathbf{W}_{\text{dec}}$, denoted as $\mathbf{d}_i$ for $i = 1, \ldots, d_{\text{hid}}$, form a dictionary of features.

SAEs tackle the problem of neuron polysemanticity by learning a set of sparsely activating features that are quantitatively more interpretable and monosemantic than other alternative approaches do (Cunningham et al., 2023; Templeton et al., 2024). In the domain of chess, SAEs effectively learn features of strategic chess concepts and board state properties (Karvonen et al., 2024).

## 4 SAE Training Details

### 4.1 Training Data

**Chess Dataset.** We use standard rated games from Lichess Database from September 2023 to November 2023, which was already consumed during the training of Maia-2, as the *corpus* to train the SAEs. We follow the same data balancing and filtering process as training Maia-2. Specifically, we balance the number of games between players of different skill levels. This balancing strategy encourages the SAEs to learn comprehensive features for better aligning players with varying skill levels. In addition, within a single game we filter valid positions following the same procedure in McIlroy-Young et al. (2020), excluding first 10 plys in opening positions and moves made under 30 seconds of remaining clock time.

**Model Activations.** The Maia-2 model is designed to adapt its chess-playing ability to different skill levels using its skill-aware attentions. The residual backbone in the early layers of Maia-2 plays a crucial role in understanding the chess boards, providing a nuanced representation that is, however, not dependent on specific skill levels. Thus, we set the focal point on the transformer blocks of the model, where its skill-aware attention modules come into play, providing us with an appropriate lens to observe how Maia-2 adjusts its understanding and decision-making based on the injected skill ratings.

For training the SAE, we focus on the residual streams of the transformer blocks in the Maia-2 model, as illustrated in Figure 1. Under the complex architecture of Maia-2, using the residual streams as ingredients has significant advantages. First, the additive nature of residual connections allows for the preservation of information from lower layers, which is especially important as Maia-2 has its residual backbones to learn general chess knowledge. Second, it provides internal representations of how the understanding evolves at different stages inside as the model incorporates skill-based information .

It is important to note that after the final residual streams, the Maia-2 model performs mean pooling across all $C_{patch}$ patchings of a board position. To maintain consistency, when training the SAEs, we also apply mean pooling to the residual streams along the dimension of patchings from the two transformer blocks before they are fed into the SAE for reconstruction.

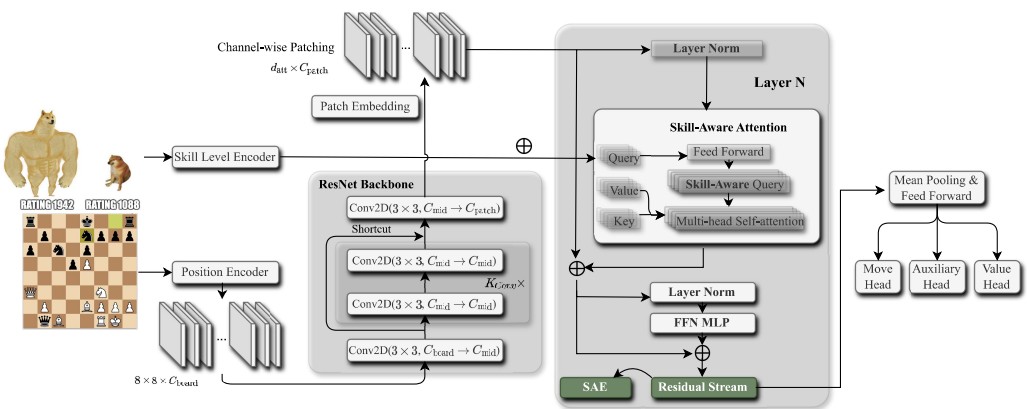

Figure 1: Locations for training SAEs in the transformer blocks of Maia-2 (coloured green).

## 4.2 CONCEPT-CALIBRATED TRAINING

Since chess as a domain carries natural sets of interpretable features and Karvonen et al. (2024) find these features effective in evaluating the quality of an SAE, we introduce a combined set of strategic concepts and board state properties to measure our training progress. Following their definition of Board State Properties (BSPs), we construct the set with 29 self-implemented strategic BSPs $\mathcal{G}_{\text{board state}}$ and 64 board state BSPs $\mathcal{G}_{\text{strategy}}$. The $\mathcal{G}_{\text{strategy}}$ comprise 32 presences of pieces at the initial position, and 32 other presences of arbitrary pieces at random squares.

To evaluate the SAE's ability to capture chess concepts, we employ the procedure as follow. Let $\mathcal{B} = \{b_1, \ldots, b_N\}$ be a set of $\mathcal{N}$ board positions, we denote $\mathcal{C} = \mathcal{G}_{\text{board state}} \cup \mathcal{G}_{\text{strategy}}$ as the set of all candidate chess concepts, where each $c \in \mathcal{C}$ is a binary classification function of $c : \mathcal{B} \to \{0, 1\}$. For each board position $b_i \in \mathcal{B}$, we compute the ground truth labels: $y_i^c = c(b_i)$ for each $c \in \mathcal{C}$, and the SAE activations $\mathbf{a}_i$, where $\mathbf{a}_i \in \mathbb{R}^d$ and $d$ is the SAE bandwidth. For each concept $c \in \mathcal{C}$ and SAE feature $j \in \{1, \ldots, d\}$, we calculate the ROC AUC score for feature over concept as:

$$\text{AUC}_{c,j} = \text{AUC}(\{y_i^c\}_{i=1}^{\mathcal{N}}, \{a_{i,j}\}_{i=1}^{\mathcal{N}}) \tag{9}$$

For each concept $c$, we take the highest AUC score across SAE features as the mastery level of this chess concept by the model. Then, we average over all candidate concepts to calculate the **CC-AUC** score, where *CC* stands for chess concept. The value of CC-AUC allows for effective calibrations and the evaluation of progress for our SAEs during training.

## 4.3 HYPERPARAMETER TUNING

**SAE Width and L1 Coefficients.** The choice of SAE hidden dimension $d_{hid}$ and L1 coefficients $\alpha$ is crucial for the SAE training, as it balances between reconstruction accuracy and learning sparse representations to disentangle features. To determine the best combination of SAE width and L1 coefficients, we employ a grid search approach. We train SAEs with $d_{hid}$ from 1x to 8x the original Maia-2 residual streams dimension, and probe $\alpha$ from 5e-6 to 1e-4. The metric used to assess the hyperparameter sweep is the sum of the $-\mathcal{L}$ and the CC-AUC score, both of which are normalized to the range of 0 to 1. By considering both the training loss and the CC-AUC score, we aim to find the hyperparameters that reach the best balance between reconstruction fidelity and the ability to capture interpretable chess concepts. For each hyperparameters set, we train the SAE using the same training data for a fixed number of 10,000 iterations to ensure a fair comparison.

**SAE Initialization.** We initialize the encoder weights and decoder weights with He Initialization (He et al., 2015) with vector in each dimension being unit-normalized. The bias vectors for encoder and decoder are set to all zeros. We choose *Relu* as the non-linear activation function.

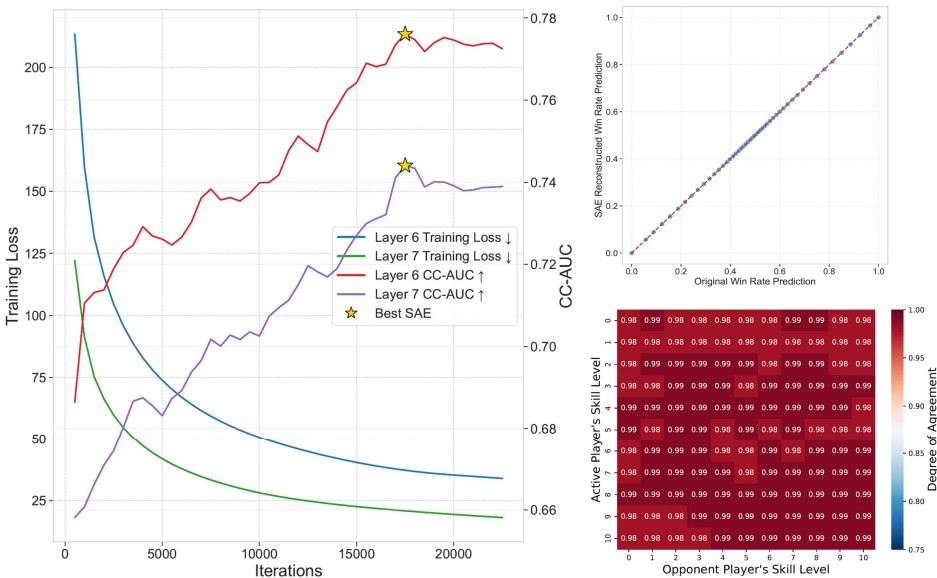

Figure 2: (*Left*) **Training progress** of sparse autoencoder on Maia 2 transformer layers residual streams. The plot shows the training loss, along with the CC-AUC scores for Layer 6 and Layer 7 of Maia 2. We use CC-AUC as a criteria for early stopping. (*Right*) **Reconstruction fidelity** of best Maia2 SAEs. In the upper half we compare the quantiles of Maia2 original and reconstructed win probability prediction. In the lower half we measure the move prediction agreement between original and reconstructed models across skill levels given identical board positions.

**Early Stopping.** By construction, the training loss of SAEs is a trade-off between reconstruction fidelity and sparsity. However, it is hard to determine the desirable stop point along the trade-off curve without external calibrations. We adopt the score of CC-AUC on the set of candidate chess concepts as our criterion for early-stopping. In practice, we leave a patience of 10 and track the CC-AUC scores with an interval of 500 training iterations.

## 5 RESULTS

### 5.1 SAE QUALITY EVALUATION

We begin our evaluation process with the hyperparameter search. In order to assess the performance of each hyperparamer setting, we use a composite metric that combines the normalized negative training loss $(-\mathcal{L})$ and the Chess Concept AUC (CC-AUC) score. Our analysis reveals that an SAE with $d_{hid} = 2048$ and $\alpha = 1e-5$ achieves the optimal balance. Detailed results of the hyperparameter sweep are presented in Appendix 7.1. To evaluate the quality of our trained sparse autoencoders, we conduct comprehensive tests focusing on two key aspects: training dynamics and reconstruction fidelity.

**Training dynamics.** The left panel of Figure 2 demonstrates the training progress of our SAEs on residual streams of Maia-2 transformer layers. We observe a consistent decrease in training loss $\mathcal{L}$ over iterations. However, the training objectives of SAEs do not support early stopping by themselves, and external calibrations are required to determine if the model gets overfitted. Following Karvonen et al. (2024), we measure the training progress of our SAEs with careful calibrations from the CC-AUC score of a set of interpretable chess concepts. Concurrently during training, we track the CC-AUC scores for the two upper transformer blocks in Maia-2. The CC-AUC scores show an initial rapid increase followed by stabilization, suggesting that our SAEs quickly learn to represent important chess features and maintain this capability throughout training. This pattern validates our use of CC-AUC as an early stopping criterion, ensuring that we halt training with a desirable balance between reconstruction fidelity and concept capturing.

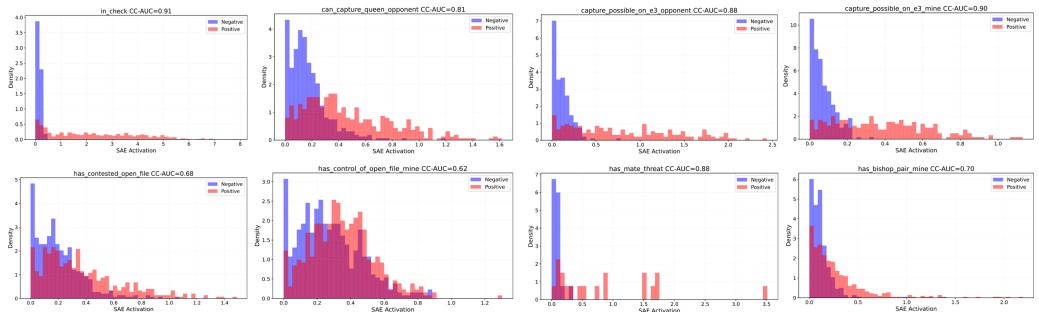

Figure 3: Maia-2 SAE's chess concept recognition measured by CC-AUC scores across a wide variety of binary concepts. Feature activations for negative samples are coloured blue and red for positive ones.

**Reconstruction Fidelity.** In the right panel, we show the high-quality reconstruction results achieved our best-performing SAEs, and we will use this set of SAEs for further experiments. In the upper half, we compare the quantiles of win probability predictions between the original Maia-2 model and our SAE reconstructions, where we replace the Maia-2 internal representations with reconstructed ones from the SAE at the training locations. The close alignment of curves across all quantiles indicates that our SAEs accurately preserve the critical information needed for win probability estimation. In the lower half, the move prediction agreements between the original and reconstructed models across identical board positions with various skill levels are examined, which further demonstrates that our SAEs faithfully reconstruct Maia-2's decision-making process with skill adaptation.

To further validate the quality of our trained SAEs, we conduct a post-hoc analysis of their ability to recognize various chess concepts. Figure 3 shows the CC-AUC scores for a diverse set of binary chess concepts, collecting the SAE's feature activations for positive and negative examples of chess positions. We note that concepts related to specific squares, such as "there is a capture possible on e3 square" for the side to play (CC-AUC = 0.88) and for the opponent (CC-AUC = 0.90), exhibit high CC-AUC scores. A comprehensive list of all strategic concepts and part of the 768 board state properties is provided in Appendix 7.1.

## 5.2 CONTEXT SPECIFIC SKILL ADAPTATION

We identify and manipulate SAE features which determine the way Maia-2 responds to threats at different skill levels. We choose threat response as our central paradigm for analyzing Maia-2's skill adaptation for three reasons. First, it is widely agreed upon that defensive skill plays a large role in separating human chess players of different strength levels. Second, we find numerous interpretable features which correspond to positions where threats exist on specific squares which Maia-2's predicted move addresses, supported by high CC-AUC scores in Figure 3 for square-level concepts. Finally, defensive features relating to individual squares are of more significance to intervene on than offensive features. In most cases there is only one option for capturing on a given square, whereas there are frequently several ways to defend or move a piece that is under attack. The selection of defensive features leaves room for Maia-2 to exercise agency within the scope of interventions that encourage defensive play focused on a specific square.

To select the SAE features which are best suited to our threat response intervention, we define a new chess concept: **square no longer attacked**. It is a binary square-level concept satisfied when at the beginning of a turn, the player has a piece on the given square which is attacked more times than defended, but after the player's move there is no piece on the square which is attacked more times than defended.

Since the residual streams of Maia-2 last layer is directly flattened and connected to its output heads, we use the last layer as the testbed for interventions. Of the 2048 SAE features, we select the one whose CC-AUC reaches highest over concept **square no longer attacked** for each square on the board. We show the AUC of the best SAE feature with each square for each **square no longer**

**attacked** concept in Figure 8 at Appendix 7.1. An example of a position where the best move satisfies **square no longer attacked** (along with the effects of our intervention on Maia-2's behavior in this position) can be found in 5

### 5.2.1 TRANSITIONAL DATASET FOR MOVE INTERVENTION

To support this investigation of skill-adaptation in Maia-2, we rely on a specially curated subset of chess positions that exhibit clear skill-dependent behavior. We start with the definition of transitional positions (Tang et al., 2024), which are positions where the model's move prediction transitions from suboptimal at lower skill levels to optimal at higher skill levels, without reverting back. Maia-2 treats 27% of all board positions monotonically, as a stark contrast with 1% in non-skill-aware models like the Maia-1 series.

We further refine this dataset to isolate positions that provide the most insight into context specific skill-based decision making. Our additional filtering criteria are as follows. First, we choose the positions where lower Elo Maia-2 makes a mistake which loses more than 30 centipawns, which frequently occurs in the transitional dataset. Second, we select positions where lower Elo Maia-2 overlooks that the specified square is currently under threat, in particular positions where the model predicts a move that does not protect the target square from attack where such a move is optimal. After the filtering process, 4750 positions remain. We perform interventions on this dataset in order to mimic the process of injecting understanding, which the player does not master or falls short in noticing, that helps significantly improve the player's move quality under common strategic types of board positions.

### 5.2.2 CAUSAL EFFECTS ON MOVE PREDICTION

**Mediated Intervention with SAE.** We propose a mediation analysis with trained SAE features for intervening on Maia-2's move prediction process. Let $\mathcal{F} = f_1, ..., f_{64}$ be the set of SAE features, where $f_i$ corresponds to the feature with the highest CC-AUC score for the **square no longer** attacked concept on the $i$-th square. For each feature $f_i \in \mathcal{F}$, we define an intervention strength $s \in \mathcal{S}$ in a predefined set ranging from 0.1 to 10. Given a chess position $p$ from our refined transitional dataset, we denote the original internal representations of model at layer $l$ as $\mathbf{x}^l \in \mathbb{R}^{d_{\text{in}}}$. We then perform intervention as:

$$\mathbf{c}^l = \sigma(\mathbf{W}_{\text{enc}}\mathbf{x}^l + \mathbf{b}_{\text{enc}}) \tag{10}$$

$$\tilde{\mathbf{c}}^l = \mathbf{c}^l + s \cdot \epsilon \cdot \mathbf{e}^i \tag{11}$$

$$\tilde{\mathbf{x}}^l = \mathbf{W}_{\text{dec}}\tilde{\mathbf{c}}^l + \mathbf{b}_{\text{dec}} \tag{12}$$

where $\epsilon$ is a small constant, and $\mathbf{e}_i$ is the unit vector corresponding to the $i$-th SAE feature. The intervened representation $\tilde{\mathbf{x}}_l$ is then used to replace the original representation in the forward pass of Maia-2 to generate outputs.

To evaluate the effect of our intervention, we compare the move predictions of the original and intervened models across different Elo ratings $E = e_1, ..., e_n$. For each Elo rating $e_j \in E$, we compute the best move prediction rate:

$$R(e_j, s, f_i) = \frac{1}{|P|} \sum_{p \in P} \mathbb{I}[\pi(\tilde{\mathbf{x}}_l, e_j) \in B(p)] \tag{13}$$

for every position $P$ in our filtered transitional dataset. $\mathbb{I}$ denotes the indicator function, and $\pi(\tilde{\mathbf{x}}_l, e_j)$ is the move predicted by the intervened model at Elo rating $e_j$, and $B(p)$ is the set of best moves for position $p$ as determined by Lichess Online Database [1]. To quantify the overall effect of our intervention, we calculate the average deviation in transition points:

---

[1]`https://database.lichess.org/#evals`

$$D(s, f_i) = \frac{1}{|P|} \sum_{p \in P} (T(p) - \tilde{T}(p)) \tag{14}$$

where $T(p)$ and $\tilde{T}(p)$ are the transition points, defined as the categorical value of the skill ratings where Maia-2 firstly transitions to optimal moves without reverting back. We also conduct a control experiment using random interventions, where we randomly select an SAE feature to manipulate instead of using the most relevant SAE feature. This allows us to distinguish between the effects of our targeted mediation and general perturbations to model internal representations.

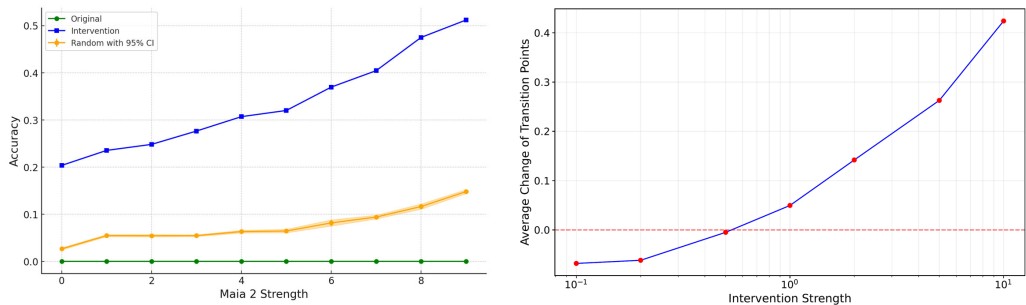

Figure 4: (*Left*) Predict Accuracies by positively boosting best SAE feature for interventions for positions where the original model makes suboptimal moves. The original accuracy (green) is 0% by the definition of transitional positions. (*Right*) Average change in transition points relative to intervention strength. Positive values in transition points indicate that the intervention causes the model to make optimal moves at lower skill levels compared to the original model.

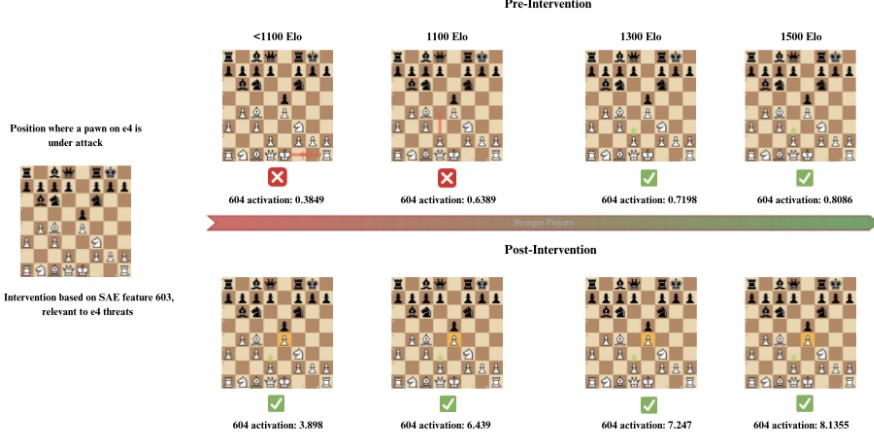

Figure 5: Diagram illustrating the effect of the SAE feature boosting intervention using a real example involving a threat on the e4 square. The correct move is d3, defending e4.

**Influence on Model Behaviour.**

Our mediated intervention reveals significant changes in Maia-2's decision-making process as a result as demonstrated in Figure 4. In the left panel, we observe the accuracy of best move prediction for positions where the original model's transition point is higher than the current skill level. By definition, the original model's accuracy for these positions is 0% across all skill levels shown.

However, after applying our targeted SAE feature interventions, we see a marked improvement in accuracy. This improvement is most pronounced at higher skill levels, reaching over 50% accuracy for skill bracket 8.

The stable increase in accuracy as skill level rises suggests that our interventions are more effective when combined with the model's inherent chess knowledge at higher skill levels. For comparison, the random intervention, represented by the orange line with a 95% confidence interval across 10 runs, shows a slight improvement over the original performance but remains significantly below the targeted intervention. This underscores the specificity and effectiveness of our mediated intervention process, demonstrating that the observed improvements are not merely a result of random perturbations to the model's internal representations.

Along the opposite dimension, we also show that increasing the relevant feature negatively sabotages the model's best move prediction accuracy in Figure 9 at Appendix 7.1.

The right panel of Figure 4 provides further insight into the effect of our interventions by plotting the average change in transition points relative to intervention strength. The positive values observed indicate that as intervention strength increases, the model tends to make optimal moves at progressively lower skill levels compared to its original behavior. This trend becomes more pronounced as the intervention strength increases, with a notable acceleration in the rate of change for intervention strengths above 1.

These causal influences on model behaviour collectively demonstrate the efficacy of our approach in modulating Maia-2's skill-dependent behavior with SAE features. By selectively enhancing the model activations to specific chess concepts through SAE feature manipulation, we can effectively boost the model's performance at lower skill levels in specific contexts, causing it to make decisions more characteristic of higher-skilled play.

## 6 DISCUSSION

### 6.1 LIMITATIONS

Although threat response represents an important component of chess skill, there are certainly skill-sensitive aspects of Maia-2's play that were not covered by our approach, such as positional chess and higher-level tactical pattern recognition. Our approach also only engages with the highest-AUC features, potentially failing to capture the full scope of Maia-2's mechanisms for modeling square specific threat response. Finally, our methodology is limited by SAEs' inherent limitation as a tool to find linear representations. The possibility of non-linear representations used to encode skill in chess models like Maia-2 and other transformers should be explored.

### 6.2 CONCLUSION

We apply sparse autoencoders to identify representations that Maia-2 uses to adjust the skill of its play. In particular, we find features in the autoencoder's latent space corresponding to the model's propensity to address threats on specific squares. We demonstrate the causal importance of these threat response features to Maia-2's predictions by adding and subtracting feature vectors corresponding to threat response on critical squares to Maia-2's internal activations. The success of our targeted intervention is evident from its unusually large impact on Maia-2's ability to navigate positions that require threat response (benchmarked against random interventions of the same magnitude). In short, we contribute to an ongoing thread of board game-based proof of concept mechanistic interpetability research by demonstrating sparse autoencoders' power to understand and manipulate skill adaptation in transformer models, a highly important but underexplored dimension of their behavior.

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

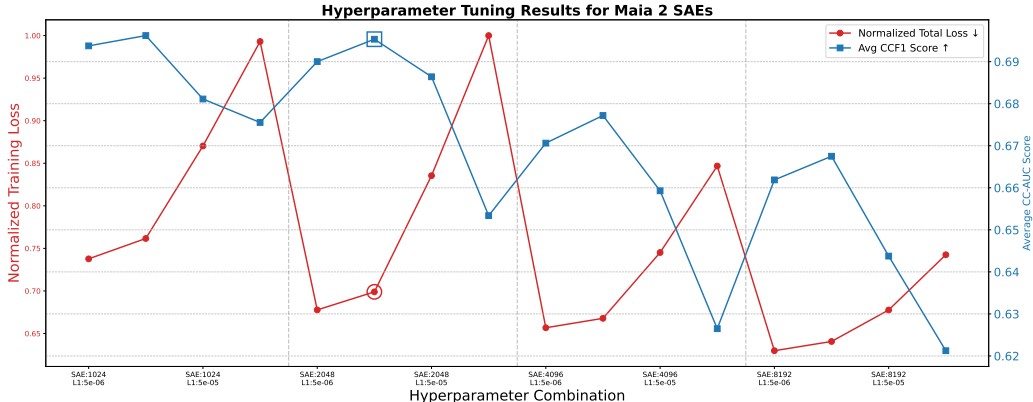

Figure 6: Hyperparameter tuning results for Maia 2 SAEs, comparing normalized total loss and average CC-AUC score across different SAE dimensions and L1 regularization coefficients.

# 7 APPENDIX

## 7.1 MORE EXPERIMENTAL RESULTS

**Hyperparameter Settings.**

We present the complete results of hyperparameter tuning for Maia 2 SAEs in Figure 6, showing the trade-off between training loss and average CCF1 score for various combinations of SAE dimensions (1024, 2048, 4096, 8192) and L1 regularization coefficients (5e-6, 1e-5, 5e-5, 1e-4). We observe that the combination of SAE dimension 2048 and L1 coefficient 1e-5, which are highlighted in Figure 6, offers an optimal balance between minimizing training loss and maximizing CC-AUC score. This configuration achieves a relatively low normalized total loss while maintaining a high average CCF1 score, indicating good model performance and generalization. Thus, we select 2048 dimensions with 1e-5 for L1 regularization as our final hyperparameters for SAE training.

**Chess Concept AUC (CC-AUC) Scores.**

Figure 7 presents a comprehensive visualization of Maia-2 SAE's chess concept recognition capabilities, as measured by CC-AUC scores for a wide range of binary chess concepts. The figure combines strategic features and board state features, showcasing the model's ability to distinguish between various chess positions and strategic elements. Each subplot represents a different chess concept named with title, with feature activations for negative samples shown in blue and positive samples in red. We also provide additional CC-AUC scores for square-specific threatens from **square no longer attacked** in Figure 8.

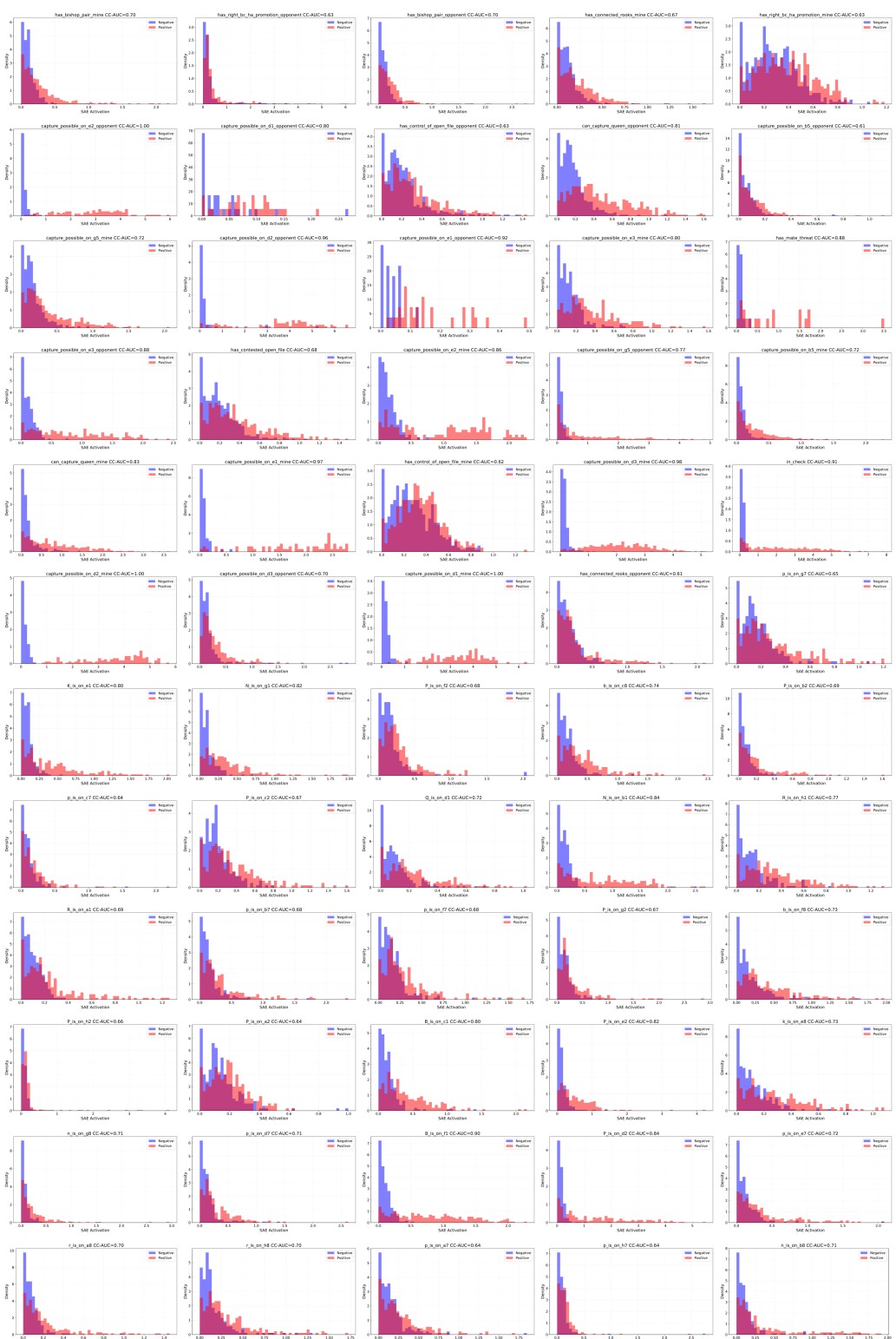

Figure 7: Maia-2 SAE's chess concept recognition measured by CC-AUC scores for all candidate binary concepts. Feature activations for negative samples are coloured blue and red for positive ones.

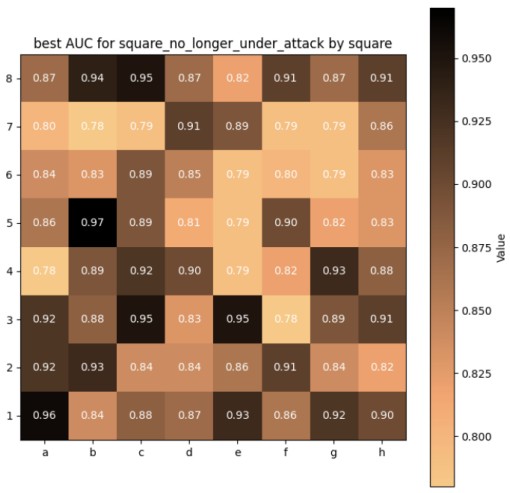

Figure 8: AUC of the best SAE feature from the second block residual stream with each square_no_longer_attacked CC, measured on 1000 randomly sampled negative examples and 200 randomly sampled positive examples

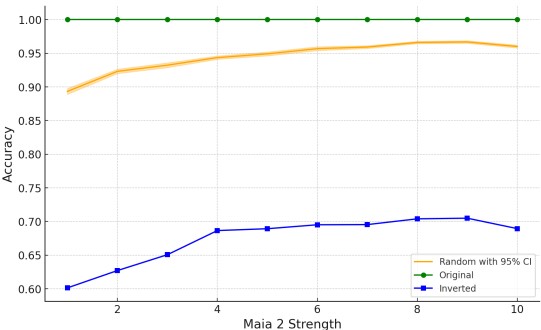

Figure 9: Accuracies by negatively boosting best SAE feature for interventions for positions where the original model makes optimal moves. The original accuracy (green) is 100% by definition.

