# OpenReview forum: "Understanding Skill Adaptation in Transformers Using Sparse Autoencoders: Chess as a Model System"
_ICLR.cc/2025/Conference — Submitted to ICLR 2025_

### Official Review · Reviewer_wbeH · 2024-10-20

**Soundness:** 2
**Presentation:** 1
**Contribution:** 1
**Rating:** 3
**Confidence:** 4

**Summary:**

The paper applies techniques for mechanistic interpretability to understanding and controlling the behavior of a chess agent. A sparse auto-encoder is trained to reconstruct predictions from an existing chess engine. Hyperparameter tuning is used to train the model to optimize for performance and capability to match a desired set of chess concepts. The SAE matches the base model closely in terms of win and move prediction.

An intervention case study modifies the last layers of the chess model using the SAE parameters selected to match a new chess concept ("square no longer attacked"). The experiments show this correctly improves (or degrades) model capacity to use this skill.

**Strengths:**

# originality
Modest.

Using mechanistic interpretability to study and intervene on networks is established as a technique, but not in chess. This application is a great fit for the method and has great potential for studying ways to shape model behavior for various desired uses.

# quality
Good.

The SAE intervention model shows very training training results (Figure 2). The changes to causally alter defensive behavior are also reasonable (to the degree they are currently present in the paper, see below). The methods for training and intervention are sound and the dataset robust.

# clarity
Modest.

Overall covers the background on Maia-2 well (perhaps too extensively) and describes the intent and structure of the experiments clearly. Some results were unclear (see below).

# significance
Good.

Mechanistic interpretability and causal model intervention is an important area of LLM research that this work directly applies. The results are potentially of broader interest due to the clear evidence of showing behavioral modification in a domain that is not about linguistic concepts, and thus relevant to transformer-based policy behavior more broadly. The hyperparameter tuning approach may be of interest to other applied domains as well.

**Weaknesses:**

# originality
The technical novelty is limited. I think the adjustments to hyperparameter tune and incorporate chess concepts for that tuning are valuable. Are there other elements that deserve greater emphasis as novel?

# quality
Only evaluating threat response makes the evaluation somewhat sparse. It would help to evaluate more concepts, perhaps by subsetting the concepts used in training and evaluating on a held-out portion of them. Right now the results are promising, but limited in scope.

# clarity
Figure 4: This seems to be the wrong figure. It is identical to Figure 8, which has a caption that makes sense for the figure. I was confused as the text describes it using different color names and patterns. This makes it hard to understand the results as the demonstration of some key results is missing.

# significance
The results would be much stronger with a human evaluation showing that modifying behavior in a desired way leads to recognizable differences in human-assessed behavior. This is beyond the scope of what could be done during reviewing, but feels like the logical way to assess that model behavioral interventions have an ecologically valid impact.

**Questions:**

Section 4.2: "where $d$ is the SAE bandwidth" - Is this $d_{hid}$?

Section 5.1: "CC-AUC scores for the two upper transformer blocks in Maia-2" - Why these two blocks? Are the other blocks broadly consistent?

Minor typos:
- "'' there is a capture" these '' should be `` for latex to produce matching quotes.
- Equation 11: $e^{i}$ should be $e_{i}$ to match the subscript notation in the text below. Might be useful to use a different letter as Elo already uses the $e$ symbol.
- Equation 13: Overload the use of the letter B is also a bit confusing. B was already used for binary concepts (the mathcal is subtle).

---

### Official Review · Reviewer_Xx5Q · 2024-10-31

**Soundness:** 2
**Presentation:** 2
**Contribution:** 2
**Rating:** 3
**Confidence:** 3

**Summary:**

This paper analyzes how transformer-based models learn a variety of skills in chess by combining Maia-2, a transformer-based model that is capable of modeling chess play across the spectrum of skill levels, and the SAE method. By training SAE, this paper identifies several chess concepts learned by Maia-2 and successfully reconstructs the prediction in Maia-2.  Besides, by intervening with the feature of the certain skill, this paper shows the potential of controlling the skill set of the chess model and thus manipulating the strength of the model or aligning the model to produce human-like behavior.

**Strengths:**

This work successfully combines the transformer-based model and the SAE technique, and shows the process of how the transformer models different skill levels. The method is simple and straightforward. By intervening in the feature in the SAE, this work successfully influenced the prediction accuracy of the model, which shows that the concept is highly related to the strength of the model.

**Weaknesses:**

1. One of the major weaknesses of this paper is its presentation. Many terminologies are not clearly defined or explained. This makes this paper hard to follow, especially the experiment section. For example, the newly discovered concepts in subsection 5.2 should be explained more clearly by adding an extra chess board for broad readers.
2. It is unclear for the design (motivation and purpose) of the experiments. In addition, the meaning of Figure 3 is not mentioned very clearly in the article. The meaning of the x-axis and y-axis in the figure needs to be mentioned in the article. For example, In Fig. 3, how the diversity is calculated is not mentioned in the article.
3. The novelty of this paper is also limited. The proposed approach basically follows the well-known method, SAE, for interpretability. Furthermore, the experiment lacks comprehensive comparisons to other approaches.
4. The experiments are only conducted in chess games, which makes the generality of the proposed method remain unclear. It would be better to include more games to demonstrate the versatility of this method.
5. This paper only focuses on the linear representation in the transformer-based model, and the non-linear representation still needs to be discovered.
6. The difference between the proposed method and Karvonen et al. (2024) is not mentioned in the article.

**Questions:**

I would kindly ask the authors to address each point listed in the weakness. In addition, the following list some more questions.
1. In Fig.3, how is the diversity calculated?
2. In Fig.6, can concepts with low ACC scores be learned through more training procedures?
3. In Section 5.2.2, mediated intervention influences the strength of the model, is it possible to manipulate the elo rating by intervening a certain concept?
4. Can this method discover more concepts that are still not acknowledged by humans, such as a high level tactical pattern recognition?
5. In Fig.2, the acquisition of the skill is defined by the highest CC-AUC score, and the score might differ in the whole training process. Is it possible that the CC-AUC of a certain skill drops after the model learns another skill? For example, the model learned a low-level skill at a small iteration and forgot it at a large iteration to learn another high-level skill.

---

### Official Review · Reviewer_omNr · 2024-11-03

**Soundness:** 2
**Presentation:** 2
**Contribution:** 2
**Rating:** 3
**Confidence:** 2

**Summary:**

This work attempts to understand “how skill shapes decision-making in complex environments”. To answer the question, this paper applied sparse autoencoder to the modulated representations of Maio-2, to "identify latent features that learn well on interpretable chess concepts".

**Strengths:**

The question being investigated in this paper is valuable and interesting.

**Weaknesses:**

From my perspective, there are quite a few unclarities in this paper.

The approach and experiment results are mixed. To my understanding, the “Mediated Intervention with SAE” part of Section 5.2.2 should be placed in the approach part. I would recommend authors reorganize the paper, decouple the approach with experiments, and describe the approach more comprehensively. Moreover, this paper directly goes to training details after preliminaries. There should be a problem formulation or description to help readers understand what is the SAE doing.

The conclusion of this paper is not clear. There is only a discussion of limitations in the rear of the paper. I think there needs to be a summary of the insights drawn from the result.

The message the paper aims to convey from the experiment parts is unclear, some terms are not well defined, and the novelty of this paper is not well justified. Please refer to my questions for these points.

**Questions:**

1.	What is the “move prediction agreement”? Is it the accuracy?
2.	What is the message conveyed from Figure 3?
3.	What are the key differences/advantages of the proposed methods compared with ”dictionary learning with sparse autoencoders”/ “linear probes”/“activation patching”, or are they addressing different problems? Is it possible to make some comparisons with those methods?

---

### Official Review · Reviewer_S3e7 · 2024-11-03

**Soundness:** 1
**Presentation:** 1
**Contribution:** 1
**Rating:** 1
**Confidence:** 3

**Summary:**

This paper studies the internal representations of Maia-2, a human-like chess model that simulates human play across different rating levels, using Sparse Autoencoders (SAEs). By defining and utilizing the proposed CC-AUC score, a metric for evaluating the quality of SAE features, the authors claim that these features can aid in understanding chess skills, particularly in threat response scenarios, and provide insights for producing human-like behavior.

**Strengths:**

* The initial story in the abstract and introduction is interesting and reasonable.
* The experiment details and the attached code link seem to help reproduce this work.

**Weaknesses:**

* This paper seems to be not ready for submission, especially when I read section 6. I don't know why it would appear in ICLR, as it could hurt the reputation of the authors. Also, there are many typos and grammar errors, along with many instances of LLM-generated style wording. I believe this version has not been proofread after copying the LLM-polished manuscript.
    * For example, checking paragraph 1 in the introduction for a proper \citet or \citep, "human-AI alignment in Chess.", and the grammar around "Maia-2 performs the skill-aware attention mechanism to fuses skill embeddings with the position representations."

* The presentation also has much space for improvement. For instance, in section 3.2, there should be some description of what property a sparse autoencoder helps with in this application, rather than just mentioning what an SAE looks like.

* Additionally, the initial claim in the abstract and introduction seems significant, but the actual results are very specific to chess with domain knowledge. It is not clear how the methodology can be generalized to other applications.

* It is also unclear what specific problem is solved or handled by the proposed method that previous methods cannot. I do not see how intervening in the representation can guide the chess model to desired behaviors.

**Questions:**

* Are you sure you submitted the final version?

* Where is the experimental evidence to support "SAE features can help shed light on how skill-specific information is encoded within a model to produce human-like behavior, and that these insights can be applied to steer the model’s performance on downstream tasks."?

* Why do you use ⊮ as the indicator function rather than a more common indicator symbol?

* Is the prediction from Maia-2 after mediated intervention still reliable? For studying behaviors, why not use extra board game behavior measures?
    * For example:
        * Contrastive approach: https://arxiv.org/abs/2208.01366
        * Unsupervised approach: https://arxiv.org/abs/2408.06051

* To avoid misunderstanding of this paper, can you specify the core idea of this paper and which experimental results support this idea? Also, it would be helpful to specify what downstream tasks this work aims to address.

---

### Meta-Review · Area_Chair_yLBx · 2024-12-09

**Metareview:**

This paper proposes to study the latent representation of Maia-2, a human-like model for playing chess, using sparse autoencoders to uncover the relationship between human expertise and behavior (chess ratings and moves). Experiments involving evaluating the learned representations and the downstream intervention strategies are conducted.

Overall, the reviewers agree that this is an important and interesting research question. However, the main concerns relate to the presentation. The paper appears rushed and would benefit greatly from a thorough revision to clarify various aspects. We hope the authors find the reviewer comments helpful in preparing their next revision.

**Additional Comments On Reviewer Discussion:**

There is a consensus among the reviewers, as noted in the meta-review.

---

### Decision · Program_Chairs · 2025-01-22

Reject